# 27-Hydroxymangiferolic Acid Extends Lifespan and Improves Neurodegeneration in *Caenorhabditis elegans* by Activating Nuclear Receptors

**DOI:** 10.3390/molecules30051010

**Published:** 2025-02-21

**Authors:** Xiaoyan Gao, Jing Yu, Yin Li, Hang Shi, Lijun Zhang, Minglv Fang, Ying Liu, Cheng Huang, Shengjie Fan

**Affiliations:** 1School of Pharmacy, Shanghai University of Traditional Chinese Medicine, Shanghai 201203, China; gxy170402310@outlook.com (X.G.); 401325yj@shutcm.edu.cn (J.Y.); kyshihang@126.com (H.S.); lijunzhang96@163.com (L.Z.); fangminglv@163.com (M.F.); lydiaaaaa_liu@163.com (Y.L.); 2School of Agriculture and Medicine, Hebei Open University, Shijiazhuang 050080, China; liyin@hebnetu.edu.cn

**Keywords:** 27-hydroxymangiferolic acid, nuclear receptor, lifespan, detoxification, Alzheimer’s disease, Parkinson’s disease

## Abstract

27-Hydroxymangiferolic acid (27-HMA) is a naturally occurring compound in mango fruits that exhibits diverse biological functions. Here, we show that 27-HMA activates the transcriptional activity of farnesoid X receptor (FXR), a nuclear receptor transcription factor, extending the lifespan and healthspan in *Caenorhabditis elegans* (*C. elegans*). Meanwhile, the longevity-promoting effect of 27-HMA was attenuated in the mutants of *nhr-8* and *daf-12,* the FXR homologs, indicating that the longevity effects of 27-HMA in *C. elegans* may depend on nuclear hormone receptors (NHRs). Further analysis revealed potential associations between the longevity effects of 27-HMA and the insulin/insulin-like growth factor-1 signaling (IIS)/TORC1 pathway. Moreover, 27-HMA increased the toxin resistance of nematodes and activated the expression of detoxification genes, which rely on NHRs. Finally, 27-HMA improved the age-related neurodegeneration in Alzheimer’s disease (AD) and Parkinson’s disease (PD) *C. elegans* models. Taken together, our findings suggest that 27-HMA is a novel FXR agonist and may prolong lifespan and healthspan via activating NHRs.

## 1. Introduction

Age extension and reducing the occurrence of aging-related diseases have emerged as urgent demands in modern medicine. Many studies have shown that the genetic manipulation of multiple signaling pathways, pharmacological intervention, and dietary restrictions are capable of extending lifespan and healthspan in multiple animal models [1,2]. However, the application of these methods is restricted to the broader population.

Nuclear receptors (NRs) play key roles in managing vital physiological and metabolic processes, such as bile acid metabolism and glycolipid metabolism [3,4]. Evidence has shown that regulating lipid signaling molecules via nuclear receptor transcription factors can extend the lifespan of *C. elegans* [5]. McElwee et al. have demonstrated that extension of lifespan is significantly associated with the elevated transcript level of genes encoding detoxification enzymes [6], which is regulated by the nuclear hormone receptors (NHRs) in mammals, such as peroxisome proliferator-activated receptors (PPARs), pregnane X receptor (PXR), farnesoid X receptor (FXR), and vitamin D receptor (VDR) [7]. We have previously shown that targeting PXR can extend lifespan and healthspan by enhancing detoxification functions in both *C. elegans* and mice [8]. FXR has multiple physiological functions, including maintaining bile acid balance, regulating lipid metabolism, resisting stress, and combating liver fibrosis [9,10]. It has been shown that FXR directly regulates xenobiotic detoxification genes in long-lived little mice [11]. Additionally, FXR regulates bile acid metabolism, which is associated with age-related neurodegenerative diseases [12]. However, it remains unclear whether FXR plays a role in regulating the lifespan and age-related neurodegeneration in animals. Hence, we propose that the activation of FXR could be a potential strategy for extending lifespan and improving aging-related diseases.

27-Hydroxymangiferolic acid (27-HMA) is a naturally occurring compound derived from mango fruits. Pharmacological studies have revealed that the organic acids and polyphenolic compounds in mangoes have functions such as anti-inflammatory and antioxidant properties [13]. Here, we show that 27-HMA is a novel FXR agonist that may extend the lifespan and healthspan in *C. elegans* through NHR-regulated functions and may improve the pathogenesis of Alzheimer’s disease (AD) and Parkinson’s disease (PD).

## 2. Results

### 2.1. 27-HMA Is an FXR Agonist

In screening for FXR agonists from natural products, we identified six compounds, including 2-Oxokolavenol, 2-Oxokolavelool, 27-Hydroxymangiferolic acid, 27-Hydroxymangiferonic acid, and mulberrin, that significantly enhanced the transcriptional activity of FXR (Appendix A) in a dose-dependent manner (Appendix A). 27-HMA (Figure 1a) significantly enhanced the transcriptional activity of FXR in a dose-dependent manner (Figure 1b). The EC50 value for this compound was 6.693 μM (Figure 1c). To exclude the possibility that 27-HMA acts on other nuclear receptors, we assayed PPARα/γ/β and LXRβ transactivities. Specific agonists for these receptors, including fenofibrate (FA), rosiglitazone (RSG), GW7647, and T0901317, were used as positive controls. The results showed that 27-HMA did not increase the transactivities of PPARα/γ/β or LXRβ (Figure 1d), indicating that it may be a specific FXR agonist. Next, we used molecular docking as a theoretical simulation method to analyze the binding mechanisms between 27-HMA and the FXR protein to further verify that it is a ligand agonist of FXR. The docking results showed that 27-HMA interacted with the FXR protein structure in two ways. First, the compound’s skeleton bound to the hydrophobic pockets formed by the hydrophobic amino acid residues through strong hydrophobic interactions, which was the primary reason for maintaining the activity of the molecule. Second, the carbonyl oxygen of the compound acted as a hydrogen bond acceptor and formed one hydrogen bond with the hydrogen bond donor of the His291 side chain. One of its hydroxyl groups simultaneously acted as both a hydrogen bond donor and an acceptor, forming two hydrogen bonds with Tyr358 and His444 (Figure 1e), indicating that the compound may directly bind to FXR protein. Together, 27-HMA could be a specific FXR agonist.

### 2.2. 27-HMA Extends Lifespan and Enhances Healthspan in C. elegans

To examine the impact of 27-HMA on the lifespan of *C. elegans*, N2 wild-type nematodes were treated with various concentrations (0, 20, 50, 100, 150, and 200 μM) of 27-HMA at the L4 stage. As shown in Appendix A and Appendix A, 27-HMA extended the lifespan of *C. elegans* by 7.34%, 12.78%, 16.67%, 9.41%, and 6.03% at concentrations of 20, 50, 100, 150, and 200 μM, respectively. The optimal effect was observed at a concentration of 100 μM (Figure 2a). Since dietary restriction (DR) is a known method for prolonging the lifespan of *C. elegans* [14], the *eat-2* (*ad1116*) mutant, which mimics dietary restriction, was included as a positive control (Figure 2a). *Escherichia coli* (*E. coli*) OP50 is a common food source for *C. elegans* in laboratory cultures [15]. To exclude the possibility that 27-HMA inhibits the growth of *E. coli* OP50, thereby reducing the food intake of nematodes, we measured the effect of 27-HMA on *E. coli* OP50 growth. As shown in Figure 2b, 27-HMA did not affect bacterial growth. Additionally, nematodes did not avoid bacteria mixed with 27-HMA (Appendix A). These results suggest that 27-HMA may extend the lifespan in *C. elegans*.

*C. elegans* exhibit age-related declines in behavioral and physiological activities, including lower pharyngeal pumping rate, body bends rate, and motion ability [16,17]. We then evaluated the effects of 27-HMA on these activities in *C. elegans*. The results showed that both the pharyngeal pumping rate and body bend rate were significantly improved by 27-HMA treatment compared to those in the control group (Figure 2c,d), while 27-HMA did not affect reproductive ability (Appendix A). We also observed enhanced motility in 27-HMA-treated nematodes (Figure 2e). These data indicate that 27-HMA can improve the healthspan of *C. elegans* during aging.

### 2.3. The Longevity Effect of 27-HMA Is Mediated by Nuclear Receptors in C. elegans

In *C. elegans*, NHR-8 and DAF-12 are homologs of mammalian nuclear hormone receptors [18]. NHR-8 is essential for longevity under dietary restriction, and a DAF-12 agonist can extend the lifespan of *glp-1/daf-36* mutants, indicating that both NHR-8 and DAF-12 are crucial for longevity [19,20]. Moreover, it is noteworthy that the endogenous ligand of DAF-12, dafachronic acid (DA), can bind to DAF-12 and activate its transcriptional activity, thereby influencing lifespan [5]. Similarly, could 27-HMA also regulate lifespan by interacting with nuclear hormone receptors, such as DAF-12? We investigated whether the lifespan-extending effects of 27-HMA depend on nuclear receptors in nematodes by using *nhr-8* and *daf-12* mutants. The results showed that 27-HMA did not extend the lifespan of *nhr-8* mutants (*tm1800*) and *daf-12* mutants (*rh61rh411*) (Figure 2g,h and Appendix A). These data suggest that the lifespan-extending effects of 27-HMA rely on the functions of nuclear hormone receptors NHR-8 and DAF-12.

### 2.4. Lifespan-Extending Effect of 27-HMA Is Related to Insulin/Insulin-like Growth Factor-1 Signaling (IIS) Pathway

The IIS pathway is crucial in regulating metabolism, development, reproduction, and stress resistance in *C. elegans* [21]. Inhibition of this pathway has been shown to extend lifespan in various model organisms, partly through the conserved FOXO transcription factor [22] (Appendix A). In *C. elegans*, mutations in the *daf-2* gene, homologous to mammalian IGF, result in a long lifespan in nematodes, requiring the FOXO homolog DAF-16 [21]. The *daf-2* and *daf-12* interact to regulate lifespan, and the *daf-16* and *daf-12* interaction can influence cholesterol levels [23,24]. Lifespan experiments on *daf-2* mutants (*e1370*) and *daf-16* mutants (*mu86*) showed that treatment with 27-HMA did not extend their lifespan (Figure 3b,c and Appendix A), indicating a reliance on IIS signaling. Moreover, we confirmed that the *daf-2* mRNA expression decreased significantly after 27-HMA treatment, while *daf-16* increased compared to the vehicle control (Figure 3i). 27-HMA also induced the nuclear translocation of DAF-16::GFP (Figure 3j).

Next, we examined other key molecules from the IIS pathway, including *age-1*, the first longevity gene discovered in *C. elegans*, *skn-1*, which regulates antioxidant enzymes, and *hsf-1*, which regulates heat shock proteins [25,26,27]. We found that 27-HMA did not extend the lifespan of *age-1*(*hx546*), *skn-1*(*tm4241*), or *hsf-1* mutants (*ps3651*), further suggesting its dependence on IIS signaling (Figure 3d–f and Appendix A).

The IIS pathway is also involved in the stress resistance response of nematodes, and lifespan extension is often associated with enhanced stress resistance [21]. As depicted in Figure 3g,h and Appendix A, 27-HMA treatment did not affect survival under heat stress (Figure 3g) but increased survival by 12.6% under oxidative stress (Figure 3h) and reduced malondialdehyde (MDA) levels (Figure 3k). The ABTS cation scavenging assay showed that 27-HMA did not exhibit antioxidant capacity compared to the positive control glutathione (GSH) (Figure 3l), suggesting that it enhances endogenous antioxidant capacities. We also confirmed that 27-HMA upregulated the expression of genes related to anti-stress responses, including *hsp-16.2, sod-1, sod-2, sod-3,* and *cat-1* (Figure 3m), which play important roles in the aging process [28,29]. These findings suggest that the lifespan extension of 27-HMA depends on IIS signaling and may be related to enhanced stress resistance.

### 2.5. TORC1 Pathway May Contribute to the Lifespan-Extending Effect of 27-HMA

FXR plays a crucial role in cholesterol and lipid metabolism [30]. The lipid signaling pathway promotes longevity by regulating transcription factors [5]. The TORC1 pathway integrates signals from growth factors, environmental stress, and nutrients to regulate processes such as protein synthesis, fat synthesis, and autophagy [31,32]. It is regulated by cellular energy status: when nutrients are low, AMPK is activated, leading to TORC1 inhibition and promoting anti-aging effects [33] (Appendix A).

In *C. elegans*, *let-363* senses nutrients and growth factors, while *raga-1* and *aak-2*, homologs of RagA and AMPK, play roles in improving the locomotion of aged nematodes and mediating lifespan extension under glucose restriction [34,35]. The lifespan-extending effects of 27-HMA were absent in *let-363* (*ok3018*), *raga-1*(*ok386*), and *aak-2* mutants (*ok524*) (Figure 4a–d and Appendix A), as well as in *eat-2* mutants (*ad1116*) (simulating dietary restriction) and *sir-2.1* mutants (*ok434*) (SIRT1 homolog) (Figure 4e,f and Appendix A) [36,37]. Additionally, 27-HMA also failed to extend the lifespan of *fat-2 (wa17)* and *fat-3 (ok1126)* mutants (Figure 4g,h and Appendix A), essential for polyunsaturated fatty acid (PUFA) synthesis, which regulates lipid and energy homeostasis [38].

Mitochondria are essential for energy homeostasis, with dysfunction linked to aging [39]. By observing the integrity of mitochondria in muscle cells of *Myo-3p::TOMM-20::mKate2::HA* (*foxSi16*) nematodes, we found that in young nematodes, 27-HMA did not affect mitochondrial contents, but it improved mitochondrial integrity in aged nematodes (Figure 4i,j), suggesting that 27-HMA may enhance mitochondrial homeostasis. Taken together, these data indicate that the TORC1 pathway may be involved in the lifespan-extending effect of 27-HMA, with lipid metabolism and energy homeostasis playing significant roles.

### 2.6. 27-HMA Activates Xenobiotic Detoxification via Nuclear Receptors in C. elegans

The regulation of detoxification genes is governed by evolutionarily conserved transcription regulators [7]. To investigate whether the lifespan-extending effect of 27-HMA in *C. elegans* is linked to xenobiotic detoxification, we exposed nematodes to toxins, including colchicine, chloromethyl mercury (MeHgCl), and paraquat. As shown in Figure 5a–c, 100 μM 27-HMA reduced mortality rates caused by all three toxins in N2 nematodes. Conversely, mortality rates in *daf-12* mutants *(rh61rh411)* and *nhr-8* mutants *(tm1800)* did not change significantly after 27-HMA treatment (Figure 5a–c and Appendix A), suggesting that *daf-12* and *nhr-8* are critical factors in 27-HMA-induced toxin resistance in *C. elegans*.

To further assess the detoxification effects of 27-HMA, we conducted qRT-PCR assays to examine the expression of detoxification genes. Treatment with 27-HMA upregulated the expression of genes, including *cyp13a7, cyp14a1, cyp35a5, cyp35b1, cyp35b2, cyp35b3, gst-4, ugt-44, pgp-3, pgp-12,* and *pgp-14* in nematodes, while no significant differences were observed in the expression of other genes (Figure 5d). These findings collectively suggest that the longevity-promoting effects of 27-HMA may be mediated through its enhancement of detoxification functions facilitated by nuclear hormone receptors NHR-8 and DAF-12.

### 2.7. 27-HMA Alleviates the Paralysis of AD in C. elegans Through Nuclear Receptors

Aging-related characteristics such as energy metabolism disorders and mitochondrial dysfunction increase susceptibility to neurodegenerative diseases [40]. FXR regulates bile acid balance, which is implicated in Alzheimer’s disease (AD). Analysis of bile acid profiles and AD markers revealed that bile acid ratios, such as glycochenodeoxycholic acid (GDCA)/cholic acid (CA), taurodeoxycholic acid (TDCA)/CA, and glycocholic acid (GLCA)/chenodeoxycholic acid (CDCA), negatively correlate with Aβ_1-42_ levels in cerebrospinal fluid (CSF), while glycochenodeoxycholic acid (GCDCA), GLCA, and taurolithocholic acid (TLCA), positively correlate with phosphorylated tau protein [12].

Given the lifespan-extending effects of 27-HMA in *C. elegans* mediated by nuclear receptors, we further explored its potential roles in ameliorating the pathogenesis of AD. Aβ deposition is a hallmark pathological feature of AD [41]. 27-HMA delayed paralysis in transgenic *C. elegans* CL4176 (Figure 6a and Appendix A), which express human Aβ_1-42_ [42], and in CL2120, which exhibits progressive paralysis [43]. Both strains showed a delayed onset of paralysis following 27-HMA treatment (Figure 6b and Appendix A). Furthermore, analysis of PT50, the time at which half of the nematodes were paralyzed, demonstrated delayed paralysis onset in both CL4176 and CL2120 nematodes treated with 27-HMA (Figure 6c,d).

To investigate whether the effect of 27-HMA on CL4176 paralysis is mediated through the nuclear receptors, we employed RNA interference (RNAi) to silence *nhr-8* and *daf-12* in CL4176 nematodes. As shown in Figure 6e–g and Appendix A, CL4176 nematodes fed with L4440 exhibited delayed paralysis, while the protective effects of 27-HMA were attenuated in nematodes treated with *nhr-8* RNAi and *daf-12* RNAi. Correspondingly, PT50 values were markedly higher in L4440-fed CL4176 nematodes, with no significant changes observed following *nhr-8* RNAi or *daf-12* RNAi treatment (Figure 6h–j). Additionally, qRT-PCR analysis revealed upregulation of mRNA for *nhr-8*, *hsp-16.2*, *sod-2*, *pgp-3*, *pgp-12*, *pgp-13*, *gst-4,* and *gst-10* in CL4176 nematodes (Figure 6k). Taken together, 27-HMA may alleviate paralysis in both CL4176 and CL2120 transgenic *C. elegans* models, likely mediated via nuclear hormone receptors NHR-8 and DAF-12, offering potential therapeutic benefits for AD.

### 2.8. 27-HMA Improves the Pathology and Behavior of PD in C. elegans

Parkinson’s disease (PD) ranks among the most prevalent neurodegenerative diseases affecting the elderly. Recent studies have highlighted the pivotal role of nuclear receptor Nurr1 in the progression of dopaminergic neurons, positioning nuclear receptor activation as a promising therapeutic avenue for PD treatment [44]. The neurotoxin 6-hydroxydopamine (6-OHDA) selectively damages dopaminergic neurons, inducing PD-like symptoms [45]. Transgenic *C. elegans* UA57, which overexpresses CAT-2 protein, facilitates dopaminergic neuron degeneration [46]. We induced N2 and UA57 nematodes with 6-OHDA to accelerate and amplify dopaminergic neuronal degeneration [47] and evaluated their locomotion via body bends. The results showed that akin to the positive control cabergoline (Caber), 27-HMA reduced 6-OHDA-induced deficits in N2 body bends both in the presence and absence of *E. coli* OP50 bacteria (Figure 7a,b). Similarly, UA57 body bends were improved in the absence of *E. coli* OP50, while no significant difference was noted in its presence (Figure 7c,d).

Furthermore, the loss of dopaminergic neurons is a hallmark of PD [44]. Assessing dopaminergic neurons labeled with GFP protein in the heads of 6-OHDA-induced UA57 nematodes, we observed a trend toward increased fluorescence intensity in the 27-HMA-treated group, although the difference was not statistically significant compared to the vehicle (Figure 7e,f). Additionally, pathological aggregation of α-synuclein is implicated in dopaminergic neuron decline [46]. To further validate the effects of 27-HMA on α-synuclein, we assayed NL5901 nematodes, a PD model expressing human α-synuclein [48]. The results showed that 27-HMA treatment attenuated pathological α-synuclein aggregation (Figure 7g,h). In summary, these results suggest that 27-HMA may exert beneficial effects on the pathology and behavior of PD in *C. elegans*.

## 3. Discussion

In this study, we demonstrated that 27-HMA, a component in mango fruits, can extend the lifespan and healthspan in *C. elegans* via NHR-dependent functions. 27-HMA increased the survival time of *C. elegans* and improved behavioral and physiological activities, such as pharyngeal pumping rate, body bend rate, and motility in aged nematodes. Several lines of evidence indicate that the longevity effects of 27-HMA are linked to NHR signaling. First, we showed that 27-HMA is a novel FXR agonist using reporter gene analysis and molecular docking assay. Second, the longevity effects of 27-HMA were attenuated in the mutant lines of NHR-8 and DAF-12, the homologs of mammalian nuclear receptors in *C. elegans*. Third, the expression of NHR downstream genes was significantly upregulated by 27-HMA. FXR has been proven to alter the expression of xenobiotic detoxification genes to increase stress resistance in long-lived little mice [11], while NHR-8 and DAF-12 also regulate the expression of detoxification genes [7]. Thus, we believe that the age-extending effects of 27-HMA may rely on the activation of NHRs.

Interestingly, we found that the longevity effects of 27-HMA were due to the enhancement of detoxification functions. 27-HMA upregulated the expression of detoxifying enzymes and transporters and increased the survival rates of nematodes when exposed to toxicants. However, the longevity effects were attenuated in *nhr-8* and *daf-12* mutants lacking detoxification gene expression. Recently, we identified that the activation of PXR can extend the lifespan and healthspan in *C. elegans* and mice through detoxification functions [8]. It has been reported that FXR also regulates detoxification gene expression [9,10]. Thus, the enhancement of detoxification gene expression by 27-HMA may be due to the activation of NHRs in *C. elegans*.

In the present study, we showed that the lifespan-extending effect of 27-HMA may depend on IIS and TORC1 pathways, which effectively improve the oxidative stress of nematodes, maintaining energy homeostasis, increasing the expression of detoxification genes, and enhancing the resistance to exogenous toxins [21,31,32,33]. This could be due to the fact that NHR-8 and DAF-12 may interact with the IIS pathway [18]. *daf-12* interacts with *daf-2*, the upstream gene of the IIS pathway, to regulate lifespan, while *daf-16* and *daf-12* interactions influence cholesterol levels [23,24].

Aging is accompanied by declines in physiological functions and increases in the risks of aging-related diseases, such as AD and PD. Here, we observed that in age-related neurodegeneration models, 27-HMA delayed paralysis in AD nematodes CL4176 and CL2120 rescued the impaired motor ability of 6-OHDA-treated PD nematodes UA57 and alleviated the α-synuclein deposition in nematodes NL5901, suggesting that 27-HMA may also improve the pathogenesis and behaviors of AD and PD nematodes. Impressively, the effects of 27-HMA were attenuated under *nhr-8* RNAi and *daf-12* RNAi treatment. The evidence supports that the improvement effect of 27-HMA on AD and PD may also be mediated by NHRs.

In conclusion, we identified that 27-HMA was a novel FXR agonist, which extended the lifespan and healthspan in *C. elegans* and improved the pathogenesis and behaviors of AD and PD. Mechanistically, the beneficial effects of 27-HMA may be achieved through the activation of nuclear hormone receptors NHR-8 and DAF-12. Our results suggest that FXR agonist is likely a feasible strategy for longevity and health promotion.

## 4. Materials and Methods

### 4.1. Materials

27-HMA (CAS No. 17983-82-3) was obtained from BioBioPha Co., Ltd. (Kunming, China) with purity ≥ 98.0% and dissolved in dimethyl sulfoxide (DMSO) to make a 10 mM stock solution. Work solutions were diluted in *E. coli* OP50, resulting in final concentrations at 20, 50, 100, 150, and 200 μM. The final concentration of DMSO was 1‰.

### 4.2. Cell Cultures and Dual-Luciferase Reporter Assays

HEK 293T cells (CRL-11268, ATCC, Manassas, VA, USA) were seeded on 48-well plates (2 × 10^5^ cells/well) and grown to 50–80% confluence with high-glucose DMEM containing 10% fetal bovine serum (FBS, Hyclone, Logan, UT, USA) at 37 °C in 5% CO_2_. The reporter gene assay was performed using a Dual-Luciferase Reporter Assay kit (Promega, Madison, WI, USA). The transfection efficiencies were normalized according to pREP7 (Renilla luciferase) activity. The expression plasmids phFXR, phRXR, FXR-dependent reporter (EcRE-Luc), or pCMXGal-hPPARα,β,γ LBD, LXRβ LBD, and the Gal4 reporter vector MH100 × 4-TK-Luc were co-transfected with a reporter construct so that 1 μg of the relevant plasmids combined with 1 μg of reporter plasmids and 0.1 μg of pREP7 reporter per ml of DMEM. After 24 h, 27-HMA and PPARα, PPARβ, PPARγ, LXRβ agonists were added to fresh media, and the cells were incubated for another 24 h to determine luciferase activity.

### 4.3. Molecular Modeling Assay

Molecular modeling and docking assays were conducted using the Molecular Operating Environment (MOE) software, version 2008.10 (Chemical Computing Group, Montreal, QC, Canada). The crystal structure of NR FXR (PDB code 1OT7) was retrieved from the Research Collaboratory for Structural Bioinformatics Protein Data Bank (PDB). All water molecules in PDB files were removed, and hydrogen atoms were subsequently added to the protein. The compound was built using the builder interface of the MOE program, and energy was minimized using the MMFF94x forcefield. Then, the compound was docked into the active site of the protein by the “Triangle Matcher” method, which generated poses by aligning the ligand triplet of atoms with the triplet of alpha spheres in the cavities of tight atomic packing. The Dock scoring in MOE software was determined using the ASE scoring function, and Forcefield was selected as the refinement method. The best 10 poses of molecules were retained and scored. After docking, the geometry of the resulting complex was studied using MOE’s pose viewer utility.

### 4.4. Strains and Maintenance Conditions

All nematodes were cultured at 20 °C, except *fat-2,* which was maintained at 25 °C, and CL4176 and CL2120 were maintained at 16 °C on a solid nematode growth medium (NGM) plate with *E. coli* OP50 [15]. The *C. elegans* strains used in this study included N2 Bristol strain (wild type), AA86: *daf-12* (*rh61rh411*), *nhr-8* (*tm1800*), CB1370: *daf-2* (*e1370*), CF1038: *daf-16* (*mu86*), MAH97: *muIs109 [daf-16p::GFP::DAF-16 cDNA + odr-1p::RFP]*, VC199: *sir-2.1* (*ok434*), *skn-1* (*tm4241*), AD1116: *eat-2* (*ad1116*), GA1001: *aak-2* (*ok524*), VC2312: *let-363* (*ok3018*), VC222: *raga-1* (*ok386*), BX26: *fat-2* (*wa17*), VC788: *fat-3* (*ok1126*), SJZ47: *FoxSi16 [myo-3p::Tomm-20::mKate2::HA]*. Rollers, CL4176: smg-1(cc546ts); *dvIs27 [Pmyo-3::human Amyloid beta 1-42; let-851 3’UTR; rol-6(su1006)]*, CL2120: *Punc-54::human Amyloid beta 1-42; mtl-2::gfp*, NL5901: *pkIs2386 [unc-54p::alpha-synuclein::YFP+unc-119(+)]*, and UA57: *baIs4 [dat-1p::GFP + dat-1p::CAT-2]*. The synchronized nematode population was obtained by sodium hypochlorite treatment [49].

### 4.5. Lifespan Assays

Lifespan analysis was carried out at 20 °C, except *fat-2,* which was carried out at 25 °C [50]. When the synchronized nematodes grew to the L4 stage, they were transferred to drug-containing and blank (DMSO only) cultures; there were 40 nematodes per group. The nematodes were transferred to refreshed plates for 5 consecutive days until reproduction ceased, then removed from the petri dish every other day. The number of dead nematodes was recorded until the last nematode died. The nematode death was judged by the absence of response to a light mechanical touch with a picker. The nematodes that exhibited bagging, exploring, or crawling off the plates were not counted. All lifespans contained three biological replicates for each experiment.

### 4.6. Bacterial Growth Assay

Bacterial growth assays were performed as outlined previously [51]. *E. coli* OP50 was plotted onto solid LB medium and incubated overnight at 37 °C in inverted position. The following day, a colony was selected and inoculated into liquid LB medium and then incubated at 37 °C for 14–16 h. Then 5, 10, 20, 50, and 100 μM of 27-HMA were added to the cultured *E. coli* OP50. The samples were placed on a shaker and incubated at 37 °C for 12 h. OD595 values were measured at four different time points (0, 4, 8, and 12 h). The experiments were repeated three times independently, and the statistical significance of growth inhibition was assessed by multiple *t*-tests.

### 4.7. Chemotaxis Assay

Six small circles equidistant from the center of the circle were first drawn on an NGM plate with a diameter of 6 cm. Subsequently, 20 μM of *E. coli* OP50 bacterial containing different concentrations of 27-HMA (0, 20, 50, 100, 150, and 200 μM) was added to each of the six small circles; 50 N2 nematodes at the L4 stage were transferred to the center of the circles on NGM plates. Two hours later, the number of nematodes crawling onto each lawn was counted.

### 4.8. Progeny Production Assay

Ten synchronized N2 nematodes at the L4 stage were transferred to NMG plates with or without 100 μM 27-HMA (one nematode per plate) to count the number of progeny that hatched. Nematodes were transferred to a new dish every 24 h until spawning ceased. All plates continued to be incubated in a 20 °C incubator to record the number of eggs hatched. The experiment was repeated three times.

### 4.9. Locomotion and Pumping Rate Assays

Synchronized N2 nematodes were cultured as lifespan assays. The body bends and pharyngeal pumping rate assays were carried out on days 3, 5, 8, 10, 12, and 15 to determine the number of body twisting and pharynx swallowing events of nematodes that occurred within 30 s. At least 10 nematodes were tested in one group, and the experiments were repeated three times. The motion assays were carried out on days 9, 11, 13, and 15, and motions were classified into three groups: Motion A (moved freely), Motion B (moved slowly after stimulation), and Motion C (only moved head and tail after prodding) (*n* = 50 nematodes in each group each time).

### 4.10. DAF-16::GFP Nuclear Localization Assay

Nematodes *muIs109 [daf-16p::GFP::DAF-16 cDNA + odr-1p::RFP]* were synchronized and treated with 100 μM 27-HMA from L1 to L4, anesthetized with 0.2% NaN_3_, and mounted on 2% agarose pads. The GFP fluorescent signals of DAF-16 localization were examined in 10 animals per condition and captured by a confocal microscope (SP-8 Leica, Wetzlar, Germany). Images were acquired with a digital camera. The number of GFP-positive nuclei of each nematode was calculated.

### 4.11. Stress Resistance Assays

The stress resistance capacity assay of *C. elegans* was established according to the described protocol [52]. N2 nematodes were cultured with 100 μM of 27-HMA for 5 days and then exposed to 8 mM hydrogen peroxide (H_2_O_2_) for oxidative stress and 35 °C for heat shock. The dead nematodes were recorded every 1 h until all nematodes died. These assays were repeated three times with 60 individuals per assay.

The enzymatic activity of malondialdehyde (MDA) in nematodes was detected using an MDA kit (Biyuntian, Shanghai, China). L1-stage N2 nematodes were treated with 27-HMA until the nematodes had grown to the L4 stage and then were washed completely with M9 buffer and ground for 120 s with a High-Throughput Tissue Grinder (SCIENTZ, Ningbo, China). Nematode proteins were extracted, and the total protein in each sample was then measured by an ELISA kit based on the BCA method.

The total antioxidant capacity of 27-HMA was detected using a Total Antioxidant Capacity Assay Kit with the ABTS method (Biyuntian, Shanghai, China). The total antioxidant capacity of 27-HMA was calculated by detecting the absorbance at 414 nm using 3-ethylbenzthiazoline-6-sulfonic acid (ABTS) as the chromogenic agent. Glutathione was used as the positive control.

### 4.12. Quantitative Real-Time Polymerase Chain Reaction (qRT-PCR) Assay

L4-stage N2 nematodes treated with 100 μM 27-HMA were washed with M9 buffer, and then total RNA was isolated from more than 10,000 nematodes using Trizol reagent (Vazyme, Nanjing, China). Total RNA was then used as a template for reverse transcription into complementary DNA using a cDNA kit (Vazyme, Nanjing, China) in conjunction with ABI StepOnePlus Real-Time Polymerase Chain Reaction System (Applied Biosystems, Waltham, MA, USA). This was followed by RT-qPCR using SYBR Green PCR Master Mix (Vazyme, Nanjing, China). β-actin was used as the internal reference for the expression level of mRNA of all genes. Statistical analysis was carried out by using the 2^−ΔΔCt^ method. The sequences of all primers are listed in Appendix A.

### 4.13. Toxicity Assay

Exogenous toxic substances methylmercury chloride (MeHgCl), paraquat (PQ), and colchicine (CC) were used to test the detoxification effect of nematodes. L4-stage N2, *nhr-8,* and *daf-12* mutants were treated with or without different concentrations of 27-HMA for 6 days according to lifespan assays and then picked into 96-well microplate containing 2 μM MeHgCl, 200 mM PQ, and 4 mM CC with 12.5, 25, 50, and 100 μM 27-HMA and diluted to 1/1000. Six nematodes per well (48–60 individuals total) were used in one experiment. The nematodes were incubated for the specified time points and monitored under a stereomicroscope.

### 4.14. Mitochondrial Integrity Analysis

The *myo-3p::Tomm-20::mKate2::HA (FoxSi16)* nematodes were synchronized and treated with DMSO and 100 μM 27-HMA. The nematodes were then transferred to slides coated with a 2% agarose pad and then anesthetized with 0.2% NaN_3_. The integrity of mitochondria was observed under a German Leica laser scanning confocal microscope. At least ten nematodes were observed per group. The fluorescence intensity of somatic mitochondria was characterized using ImageJ (version 1.53k) software.

### 4.15. Neurodegenerative Disease Assays

CL4176 were exposed to 100 μM 27-HMA at 16 °C for 72 h and then transferred to 25 °C to induce Aβ expression [53]. The nematodes that only moved their head or did not show a full-body wave when touched were scored as paralyzed. Paralysis was assessed every 2 h. Similarly, CL2120 were checked daily for paralysis under the same conditions.

For RNA interference (RNAi) paralysis assays, synchronized L1-stage CL4176 were fed with *E. coli* HT115 containing an empty control vector (L4440) until L4, then transferred to plates where they were fed with *nhr-8* RNAi or *daf-12* RNAi constructs with 27-HMA. All RNAi constructs were obtained from the Ahringer RNAi library and grown at 37 °C overnight in LB-containing ampicillin (50 μg/mL) after sequence verification [54].

UA57 and N2 nematodes were grown synchronously at 20 °C to the L1 stage. Approximately 500 nematodes were then transferred to a new 1.5 mL EP tube and centrifuged at 3500 rpm for 3 min. The supernatant was discarded, and the nematodes were washed three times with M9 buffer and resuspended. The nematodes were incubated with 100 μM 27-HMA for 30 min, followed by a 1 h incubation with 6-OHDA (10 mM) in the dark, washed three times again, and transferred to new NGM plates containing 27-HMA. The blank and positive control groups were treated with DMSO and cabergoline (Caber), respectively. After 9 days, body bends were carried out over 30 s, and the dopaminergic neurons were observed under a German Leica laser scanning confocal microscope. Similarly, L1-stage NL5901 were exposed to 100 μM 27-HMA, DMSO, and Caber, respectively. Then, 7 days after dosing, the α-synuclein accumulations were observed under a German Leica laser scanning confocal microscope. The nematode head fluorescence intensities were quantified by ImageJ.

### 4.16. Statistical Analysis

Statistical analyses were performed using GraphPad Prism V.9.0.0 (La Jolla, San Diego, CA, USA) and SPSS (version 21.0). All data are presented as means ± standard error of the mean (S.E.M). Comparisons were carried out via one-way analysis of variance (ANOVA), two-way ANOVA, and two-tailed unpaired Student’s *t*-test. Differences were considered statistically significant when *p* was <0.05.

## Figures and Tables

**Figure 1 molecules-30-01010-f001:**
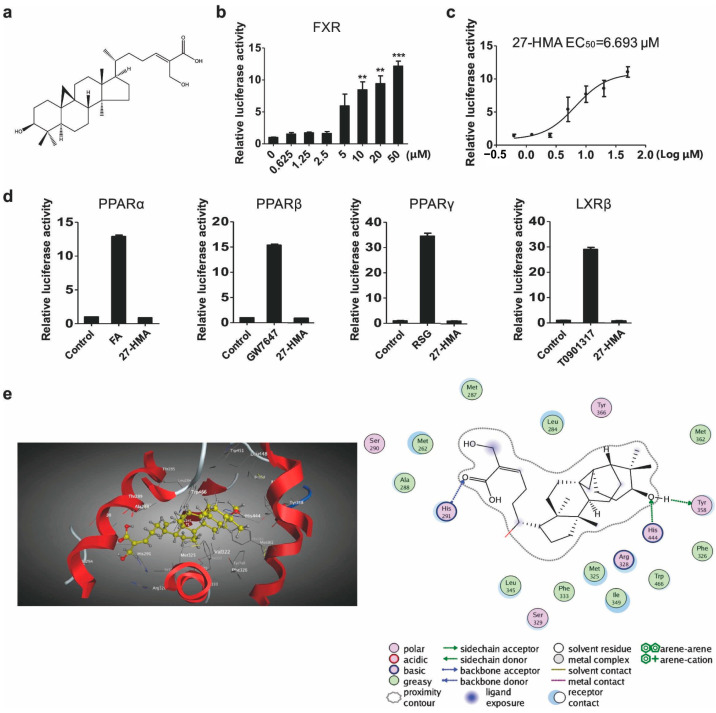
27-HMA is an FXR agonist. (**a**) Chemical structure of 27-HMA. (**b**) 27-HMA activated FXR transactivity in a dose-dependent manner, as determined by dual luciferase reporter assays. (**c**) The EC50 value of 27-HMA was 6.693 μM. (**d**) PPARα, PPARβ, PPARγ, and LXRβ transactivity. (**e**) Interactions between 27-HMA and FXR, as shown by molecular docking. All data were presented as mean ± S.E.M. Compared with vehicle group, ** *p* < 0.01, *** *p* < 0.001.

**Figure 2 molecules-30-01010-f002:**
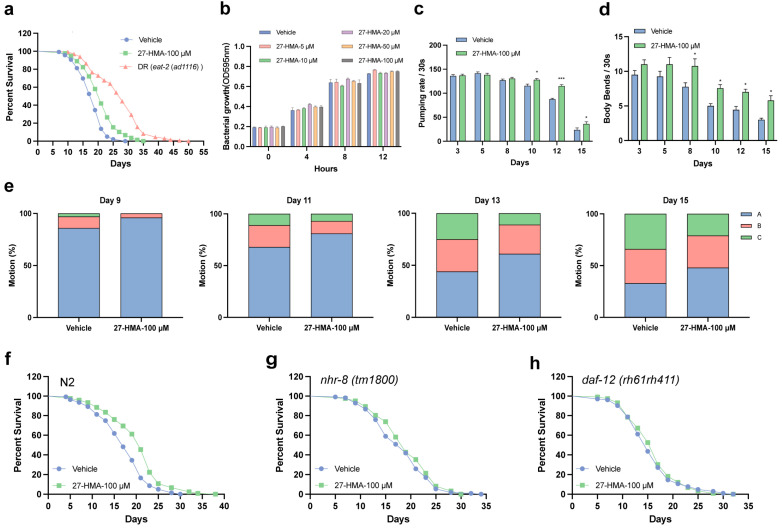
27-HMA extends lifespan and enhances healthspan in *C. elegans*. (**a**) The survival curve of wild-type nematodes treated with 100 μM 27-HMA. DR was included as a positive control. (**b**) The growth of *E. coli* OP50 with different concentrations of 27-HMA. (**c**) The pharyngeal pumping rate of nematodes treated with 27-HMA. (**d**) The body bend rate of nematodes treated with 27-HMA. (**e**) The motion of nematodes treated with 27-HMA. Motion A: moved freely; Motion B: moved slowly after stimulation; Motion C: only moved head and tail after prodding. (**f**) Survival curves of wild-type nematodes. (**g**) Survival curves of *nhr-8* mutants (*tm1800*). (**h**) Survival curves of *daf-12* mutants (*rh61rh411*). All data were presented as mean ± S.E.M. Compared with vehicle group, * *p* < 0.05, *** *p* < 0.001.

**Figure 3 molecules-30-01010-f003:**
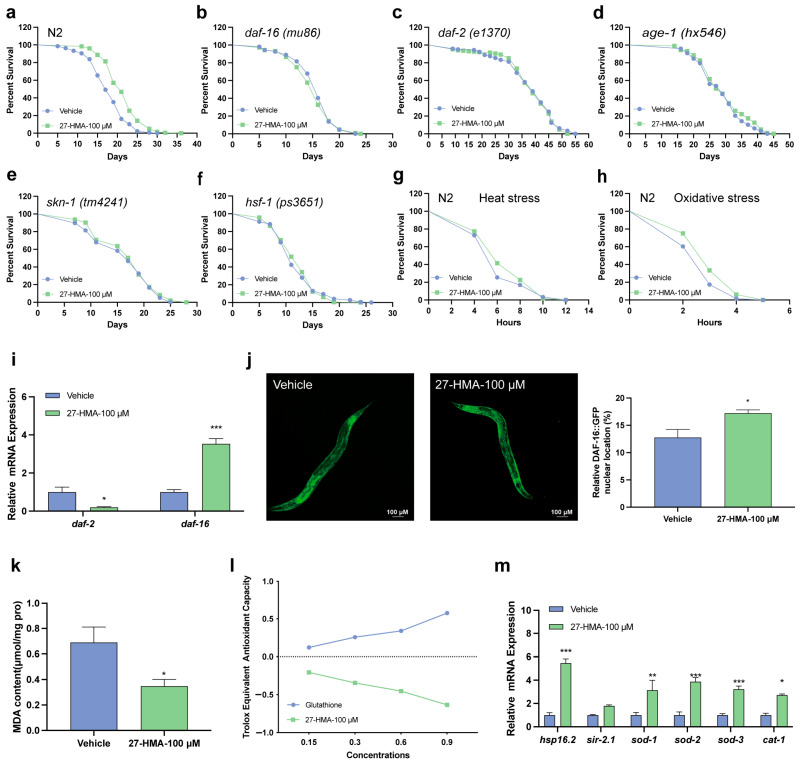
The IIS pathway is involved in 27-HMA-mediated lifespan extension. Survival curves of wild-type nematodes (**a**), *daf-16* (*mu86*) (**b**), *daf-2* (*e1370*) (**c**), *age-1* (*hx546*) (**d**), *skn-1* (*tm4241*) (**e**), and *hsf-1* mutants (*ps3651*) (**f**). (**g**) Survival curves of nematodes for heat stress. (**h**) Survival curves of nematodes for oxidative stress. (**i**) Expression of the aging-related genes *daf-2* and *daf-16* measured by qRT-PCR. (**j**) Nuclear translocation of DAF-16::GFP (*muIs109*). (**k**) MDA activity in nematodes after treatment with 27-HMA. (**l**) Trolox-equivalent antioxidant capacity in nematodes after treatment with 27-HMA. (**m**) Expression of anti-stress-related genes was measured by qRT-PCR. All data presented as mean ± S.E.M. Compared with vehicle group, * *p* < 0.05, ** *p* < 0.01, *** *p* < 0.001.

**Figure 4 molecules-30-01010-f004:**
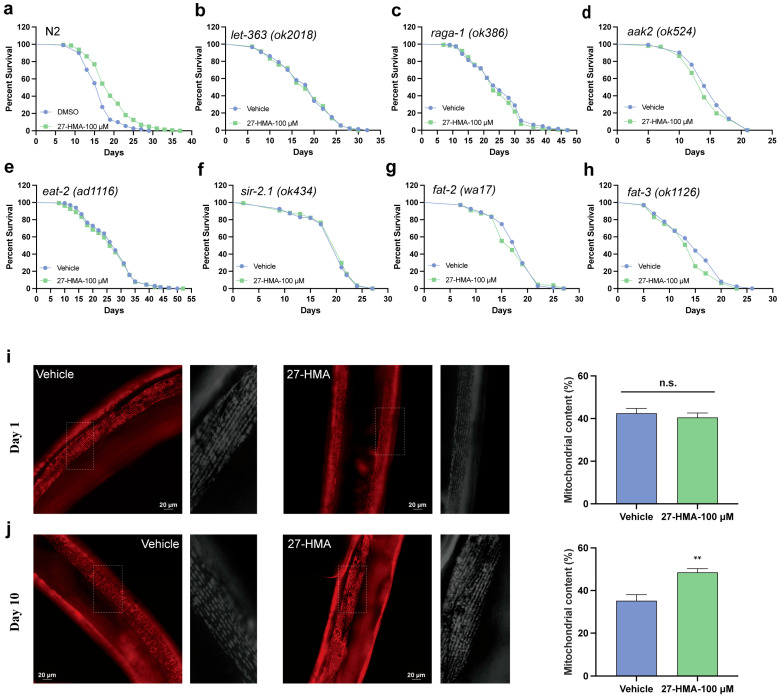
TORC1 pathway may be involved in the lifespan-extending effect of 27-HMA. Survival curves of wild-type (**a**), *fat-2 (wa17)* (**b**), *fat-3 (ok1126)* (**c**), *aak-2 (ok524)* (**d**), *let-363 (ok3018)* (**e**), *raga-1 (ok386)* (**f**), *eat-2 (ad1116)* (**g**), and *sir-2.1 (ok434)* mutants (**h**). (**i**,**j**) Representative images and RFP quantification of mitochondrial content at day 1 and day 10 of adulthood in muscle (pmyo-3 mtRFP) mito::RFP reporter strains treated with 27-HMA or vehicle. All data presented as mean ± S.E.M. Compared with vehicle group, n.s. means no significant difference, ** *p* < 0.01.

**Figure 5 molecules-30-01010-f005:**
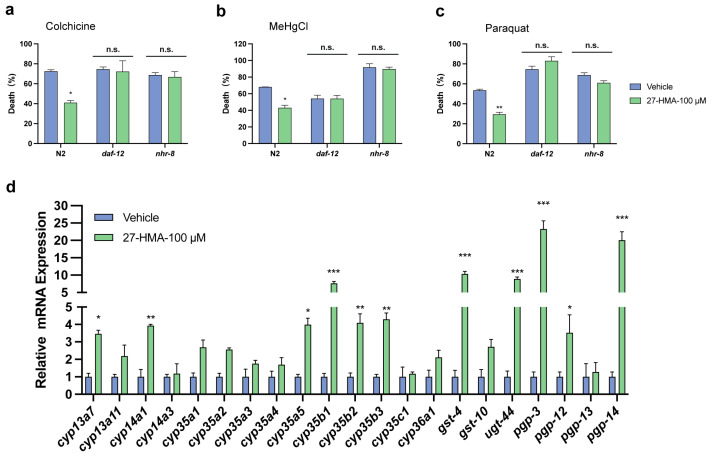
Xenobiotic detoxification may be required for 27-HMA-induced longevity. (**a**) The livability of wild-type, *daf-12 (rh61rh411),* and *nhr-8 (tm1800)* exposed to 4 mM colchicine. (**b**) The livability of wild-type, *daf-12 (rh61rh411),* and *nhr-8 (tm1800)* exposed to 2 μM MeHgCl. (**c**) The livability of wild-type, *daf-12 (rh61rh411),* and *nhr-8 (tm1800)* exposed to 200 mM paraquat. (**d**) The expression of detoxification genes was tested by qRT-PCR. All data presented as mean ± S.E.M. Compared with vehicle group, n.s. means no significant difference, * *p* < 0.05, ** *p* < 0.01, *** *p* < 0.001.

**Figure 6 molecules-30-01010-f006:**
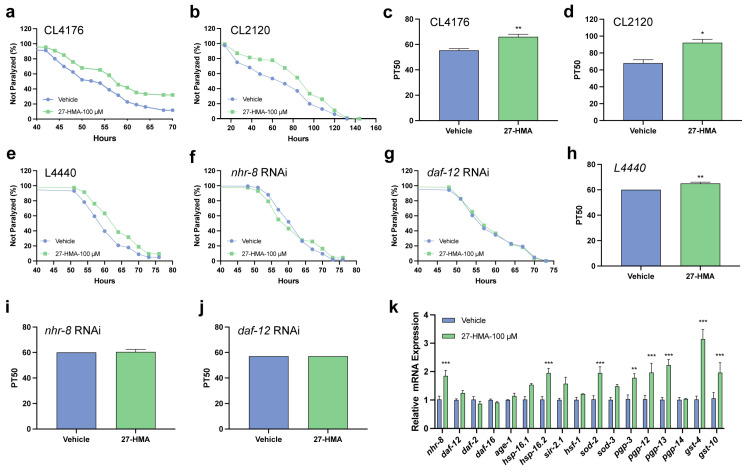
27-HMA alleviates the paralysis of AD in *C. elegans*. (**a**) Paralysis of CL4176 induced by Aβ. (**b**) Paralysis of CL2120 induced by Aβ. (**c**) PT50 of CL4176 induced by Aβ. (**d**) PT50 of CL2120 induced by Aβ. (**e**–**g**) Paralysis of CL4176 with L4440, *nhr-8* RNAi, and *daf-12* RNAi. (**h**–**j**) PT50 of CL4176 with L4440, *nhr-8* RNAi, and *daf-12* RNAi. (**k**) The expression of genes was tested by qRT-PCR in CL4176 nematodes. All data presented as mean ± S.E.M. Compared with vehicle group, * *p* < 0.05, ** *p* < 0.01, *** *p* < 0.001.

**Figure 7 molecules-30-01010-f007:**
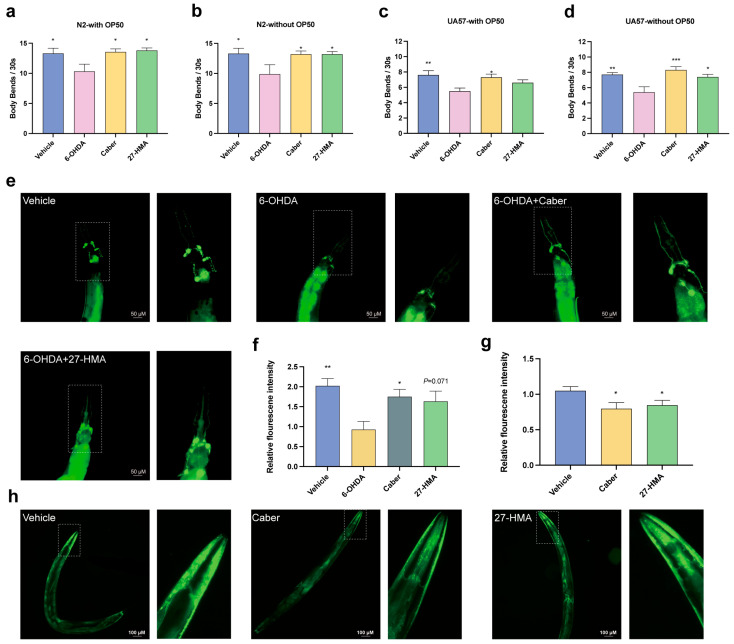
27-HMA improves the pathology and behavior of PD in *C. elegans*. (**a**,**b**) 27-HMA ameliorates 6-OHDA-induced N2 locomotion deficits. (**c**,**d**) 27-HMA ameliorates 6-OHDA-induced UA57 locomotion deficits. (**e**,**f**) Dopaminergic neurons in 6-OHDA-induced UA57. (**g**,**h**) 27-HMA ameliorates pathological α-synuclein aggregation of nematode NL5901. All data presented as mean ± S.E.M. Compared with vehicle group, * *p* < 0.05, ** *p* < 0.01, *** *p* < 0.001.

## Data Availability

All data supporting the findings of this study are available within the paper and in Appendix A.

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
