# Peer review of "27-Hydroxymangiferolic Acid Extends Lifespan and Improves Neurodegeneration in Caenorhabditis elegans by Activating Nuclear Receptors"

_molecules, 2025, doi:10.3390/molecules30051010_

Round 1

Reviewer 1 Report

Comments and Suggestions for Authors

Gao et al. manuscript drafting was excellent and well-done. Hypothesis is quite sturdy and experiments have been thoughtfully and carefully conducted, and results with conclusions have been well put out. Such work really stands out with its very much significant contribution to the field of research in C. elegans and represents a very important study carried out on subjects of aging and neuromuscular diseases.

Minor:

Figure 1 (line 61): It would be nice to include one molecule of the dafachronic acids in that figure so readers can see it compared to 27-HMA, as both are putatively part of the dafachronic acid pathway. For Panel B and the legend, note the quantification was done with minigene reporters and not with the endogenous system. Did the authors measure endogenous expression of FXR by RT-qPCR?

Figure 2 (line 88): It is really busy in Panel A; I would have to remove some of the treatments so that it is easier to interpret.

Line 99: Perhaps the text that denotes that dietary restriction extends lifespan should also be included as a positive control in Figure 2A so readers may compare its influence with that of 27-HMA.

Line 95: The culling of Supplementary Data may move this panel to Figure 2, thereby tidying the primary text.

Lines 102-103 (Figure 2G): The colors in the figure are ambiguous. The legend refers to A, B, and C, but there is no explanation for these in either the figure or its legend. Clarification is needed.

Line 105: Shall include similarity with DAF as mentioned earlier; otherwise, it is not bad to include this aspect as it would show how deep and supportive the results are.

Figure 5 (line 213): In Figures where no significant change was observed, the displays should be confined to Supplementary Data. Other than that, different concentrations of the same treatment make too much information in one figure. Usually, four concentrations for each treatment would be better to present another graph.

Author Response

Please see the attachment. Thank you very much for taking the time to review this manuscript.

Reviewer 2 Report

Comments and Suggestions for Authors

In this study, Gao et al. investigated the effects of 27-Hydroxymangiferolic acid on lifespan extension and neuroprotection in C. elegans, and explored its underlying mechanisms. The study is well-designed and executed. However, the manuscript requires substantial revisions before it can be considered for publication. Below are my specific comments and concerns.

1. Revise the gene names and allele designations to ensure they are italicized and capitalize the protein names consistently throughout the manuscript.

2. Eliminate the repetitive wording in section 2.4.

3. I recommend the authors include the strain details for all C. elegans used, for example, AA86 (daf-12(rh61rh411)).

4. I strongly suggest that the authors provide a clear and detailed explanation of all methods, including experimental conditions, treatments, replicates, biological trials, the number of individuals (e.g., worms) used, and the genotypes of the worms for each experiment.

5. I recommend the authors standardize the terminology and use "E. coli" or "E. coli OP50" throughout the manuscript.

6. The concentration of NaN3 used appears to be relatively high. A range of 10–25 mM is sufficient to anesthetize the worms.

7. I suggest the authors include appropriate references for the methods they have adopted.

Author Response

(The authors gave the same response as above.)

Reviewer 3 Report

Comments and Suggestions for Authors

Please see attached pdf. Good Luck.

Summary:
In the manuscript, authors evaluated the 27-Hydroxymangiferolic acid (27-HMA) effect in
extending lifespan and improving neurodegeneration in nematode C. elegans model of
Alzheimer's disease (AD) and Parkinson’s disease (PD). Authors demonstrated FXR
specific activity of 27-HMA, and data showed 27-HMA as FXR agonist. Authors also
showed that 27-HMA affected C. elegans life span without influencing/altering their
bacterial food (E. coli, OP50). Data showed the possible involvement of TORC1 pathway
and Insulin/insulin-like growth factor-1signaling (IIS) pathway. Authors also showed that
27-HMA improved lifespan and health span via influencing xenobiotic detoxification, and
that the 27-HMA alleviates the paralysis of AD model worms, improves the pathology and
behavior of PD model worms.
Overall, the study presented several data for the respective claims. However, the written
language and typos together makes it difficult to understand the science aptly. Hence, I
do not recommend it for publishing until it’s been revised addressing following concern in
addition to the language and typos. Please consider following comments for the revision.
Comments & concerns to be addressed:
1) Several typos: English language related editing is required, otherwise difficult to
understand the science!
2) The detail list of the FXR agonists screened is missing! Or the authors just randomly
selected 27-HMA, and clear rationale is missing!
3) Figure 1d: What is FA, RSG, GW7647, T0901317?
4) Figure 2a: Should show mean survival values with statistical significance from Table
S1, and the current curve in 2a can go in supplementary figure
5) Figure 2g: What does A, B and C stand for?
6) Define the acronyms GDCA/CA, TDCA/CA, and GLCA/CDCA.
7) If the transgenic strain UA57 expresses CAT-2 protein which facilitates dopaminergic
neuron degeneration, then what is the purpose of treating them with neurotoxin 6-hydroxydopamine
(6-OHDA)?
8) The image quality should be improved, and the scale-bars should be made legible.
9) Since there are more than 200 NHRs (nuclear hormone receptors) in C. elegans, and
authors evaluated only few of them for their potential role with 27-HMA actions,
authors should clarify this and not generalize the involvement of all NHRs.
10) The schematics of different pathways involved- for example, TORC1 pathway & IIS
pathway- showing their respective key downstream/upstream protein/genes which
were targeted via RNAi or genetic mutants will help to better visualize the rationale.!

Comments on the Quality of English Language

There are several typos and misspellings, for example "genetically manipulation" instead of "genetic manupulation", "archived" instead of "achieved" etc. Overall, it made it very difficult to understand. 

Author Response

(The authors gave the same response as above.)

Round 2

Reviewer 2 Report

Comments and Suggestions for Authors

All the best.